# Blood monocyte transcriptome and epigenome analyses reveal loci associated with human atherosclerosis

Yongmei Liu[1], Lindsay M. Reynolds [1], Jingzhong Ding[2], Li Hou[1], Kurt Lohman[3], Tracey Young[1], Wei Cui[2], Zhiqing Huang[4], Carole Grenier[4], Ma Wan[5], Hendrik G. Stunnenberg[6], David Siscovick[7], Lifang Hou[8], Bruce M. Psaty[9,10], Stephen S. Rich[11], Jerome I. Rotter[12], Joel D. Kaufman [13], Gregory L. Burke[1], Susan Murphy[4], David R. Jacobs Jr[14], Wendy Post[15], Ina Hoeschele[16], Douglas A. Bell[5], David Herrington[2], John S. Parks[17], Russell P. Tracy[18], Charles E. McCall[17] & James H. Stein[19]

Little is known regarding the epigenetic basis of atherosclerosis. Here we present the CD14+ blood monocyte transcriptome and epigenome signatures associated with human atherosclerosis. The transcriptome signature includes transcription coactivator, *ARID5B*, which is known to form a chromatin derepressor complex with a histone H3K9Me2-specific demethylase and promote adipogenesis and smooth muscle development. *ARID5B* CpG (cg25953130) methylation is inversely associated with both *ARID5B* expression and atherosclerosis, consistent with this CpG residing in an *ARID5B* enhancer region, based on chromatin capture and histone marks data. Mediation analysis supports assumptions that *ARID5B* expression mediates effects of cg25953130 methylation and several cardiovascular disease risk factors on atherosclerotic burden. In lipopolysaccharide-stimulated human THP1 monocytes, *ARID5B* knockdown reduced expression of genes involved in atherosclerosis-related inflammatory and lipid metabolism pathways, and inhibited cell migration and phagocytosis. These data suggest that *ARID5B* expression, possibly regulated by an epigenetically controlled enhancer, promotes atherosclerosis by dysregulating immunometabolism towards a chronic inflammatory phenotype.

[1] Department of Epidemiology and Prevention, Wake Forest School of Medicine, Winston-Salem, NC 27157, USA. [2] Department of Internal Medicine, Wake Forest School of Medicine, Winston-Salem, NC 27157, USA. [3] Department of Biostatistical Sciences, Wake Forest School of Medicine, Winston-Salem, NC 27157, USA. [4] Duke University, Durham, NC 27708, USA. [5] National Institute of Environmental Health Sciences, National Institutes of Health, Research Triangle Park, NC 27709, USA. [6] Department of Molecular Biology, Nijmegen Centre for Molecular Life Sciences (NCMLS), 6525 GANijmegen, The Netherlands. [7] New York Academy of Medicine, New York, NY 10029, USA. [8] Division of Cancer Epidemiology and Prevention, Northwestern University Feinberg School of Medicine, Chicago, IL 60208, USA. [9] Cardiovascular Health Research Unit, Department of Medicine, Epidemiology and Health Services, University of Washington, Seattle, WA 98101, USA. [10] Kaiser Permanente Washington Health Research Institute, Seattle, WA 98101, USA. [11] Center for Public Health Genomics, University of Virginia, Charlottesville, VA 22908, USA. [12] Institute for Translational Genomics and Population Sciences, Los Angeles BioMedical Research Institute at Harbor-UCLA Medical Center, Torrance, CA 90502, USA. [13] Department of Environmental and Occupational Health Sciences, Medicine and Epidemiology, University of Washington, Seattle, WA 98104, USA. [14] Division of Epidemiology and Community Health, School of Public Health, University of Minnesota, Minneapolis, MN 55455, USA. [15] Department of Pathology and Cardiology, Johns Hopkins University, Baltimore, MD 21205, USA. [16] Biocomplexity Institute and Department of Statistics, Virginia Tech, VA 24061, USA. [17] Department of Internal Medicine-Section on Molecular Medicine, Wake Forest School of Medicine, Winston-Salem, NC 27157, USA. [18] Department of Pathology, University of Vermont, Colchester, VT 05446, USA. [19] University of Wisconsin School of Medicine and Public Health, Madison, WI 53792, USA. Russell P. Tracy, Charles E. McCall and James H. Stein contributed equally to this work. Correspondence and requests for materials should be addressed to Y.L. (email: yoliu@wakehealth.edu)

Epigenomics and transcriptomics can illuminate the interplay between the genome and its environment and may provide insights into the molecular basis of complex diseases, including cardiovascular disease (CVD)[1–5]. Epigenetic targeting also is an attractive treatment strategy for reordering dysregulated gene expression. To date, epigenome-wide studies of CVD traits are limited[6] and their interpretation is potentially complicated by use of data from mixed cell types, which may obscure cell-type-specific functional mechanisms. Monocytes and their derived macrophages are key factors in inflammation that contribute to the development of many chronic diseases, including atherosclerosis[7–10].

We recently reported the blood monocyte epigenome and transcriptome signatures of several CVD risk factors, including age, obesity, and cigarette smoking[11–14]. In this study, we identify transcriptome and methylome features that are associated with atherosclerosis. In addition to traditional CVD risk factors, we also integrate these findings with histone modification, DNase-seq, Hi-C (genome chromosome conformation capture), and ChIA-PET (chromatin interaction analysis by paired-end tag) sequencing data and in vitro functional data, in order to characterize novel molecular mechanisms of atherosclerosis. We show emerging evidence for a potential role of ARID5B in atherogenesis, and for an epigenetically controlled regulatory site of *ARID5B* expression.

## Results

**Clinical data and sample characteristic.** The Multi-Ethnic Study of Atherosclerosis (MESA) is a multi-site, longitudinal study designed to investigate the prevalence, correlates, and progression of subclinical CVD in a population cohort of 6814 participants. Since its inception in 2000, five clinic visits have collected extensive clinical, socio-demographic, lifestyle, behavior, laboratory, nutrition, and medication data[15]. At Exam 5, carotid ultrasound and computed tomography (CT), were used to quantify carotid plaque burden (carotid plaque score, range 0–12) and coronary artery calcification (CAC Agatston score), respectively. These two measures of atherosclerosis burden independently predict future CVD events in MESA and other cohorts[16–19] (distribution of scores presented in Supplementary Fig. 1). Table 1 shows characteristics of the MESA participants at Exam 5, overall and stratified by study site (study site 1: $N = 709$, site 2: $N = 499$), including demographics (age, sex, race/ethnicity, and study site), traditional CVD risk factors (cigarette smoking, body mass index (BMI), high- and low-density lipoprotein cholesterol (HDL-C and LDL-C), hypertension, type II diabetes mellitus (T2DM)), atherosclerosis burden measures, prevalent CVD, and statin use. The transcriptome and methylome of monocytes purified at MESA Exam 5 were profiled concurrently. In a separate effort to evaluate reproducibility of single-time measures, we showed high consistency of repeated Illumina microarray data over five months (Supplementary Fig. 2), suggesting both RNA expression and DNA methylation at most loci can be stable over time in an individual.

**Blood leukocyte count and atherosclerosis.** Total white blood cell (WBC) count and its constituent subtypes were measured for all samples before monocyte purification, which provided the absolute monocyte count. Higher monocyte count, but not monocyte percentage or other leukocyte count, was marginally associated with carotid plaque score and CAC (natural log (carotid plaque score/CAC + 1), $P = 0.026$ with 0.4% of variability explained, and $P = 0.079$, respectively), in agreement with previous reports[20]. We used positive immunoselection (magnetic beads)[21] to produce samples of monocytes with >90% purity. Residual contamination with neutrophils, B cells, T cells, and natural killer cells was estimated as previously reported[22]. Likewise, the generally small percentage of CD14+ monocytes that are also CD16+ (~10%)[23] was estimated based on expression of *FCGR3a (CD16a)*. Neither the surrogates of residual cell

---

**Table 1 Population characteristics at MESA Exam 5**

| Characteristics[a] | Overall (*N* = 1208) | By site | |
|---|---|---|---|
| | | Study 1 (JHU + CO) (*N* = 709) | Study 2 (UMN + WFU) (*N* = 499) |
| *Demographic* | | | |
| Age (years) | 70 ± 9 | 70 ± 9 | 69 ± 9 |
| Sex (% female) | 51.4 | 55.0 | 46.3 |
| *Race/ethnicity* | | | |
| African American (%) | 21.5 | 36.1 | 0.8 |
| Hispanic (%) | 32.6 | 29.3 | 37.3 |
| Caucasian (%) | 45.9 | 34.6 | 61.9 |
| *Cigarette smoking* | | | |
| Former (%) | 51.7 | 49.1 | 55.5 |
| Current (%) | 9.1 | 8.1 | 10.5 |
| *Clinical and laboratory* | | | |
| BMI, (kg m$^{-2}$) | 30 ± 6 | 30 ± 6 | 30 ± 5 |
| LDL-C (mg dl$^{-1}$) | 105 ± 32 | 107 ± 33 | 102 ± 30 |
| HDL-C (mg dl$^{-1}$) | 54 ± 16 | 57 ± 17 | 50 ± 13 |
| Impaired glucose tolerance (%) | 45.6 | 44.3 | 47.5 |
| Diabetes (%) | 23.0 | 25.2 | 19.8 |
| Hypertension (%) | 60.9 | 63.8 | 56.6 |
| Carotid plaque score, median (IQR) | 2 (0–4) | 2 (0–3) | 2 (0–4) |
| Coronary artery calcium score, median (IQR) | 46 (0–305) | 32 (0–227) | 76 (0–397) |
| Prevalent CVD (%) | 5.3 | 4.2 | 6.8 |
| Statin use (%) | 38.5 | 35.7 | 42.5 |

BMI body mass index, CO Columbia University, CVD cardiovascular disease, HDL-C high-density lipoprotein-cholesterol, IQR interquartile range, JHU John Hopkins University, LDL-C low-density lipoprotein-cholesterol, UMN University of Minnesota, WFU Wake Forest University
[a]Plus-minus values are means ± SD

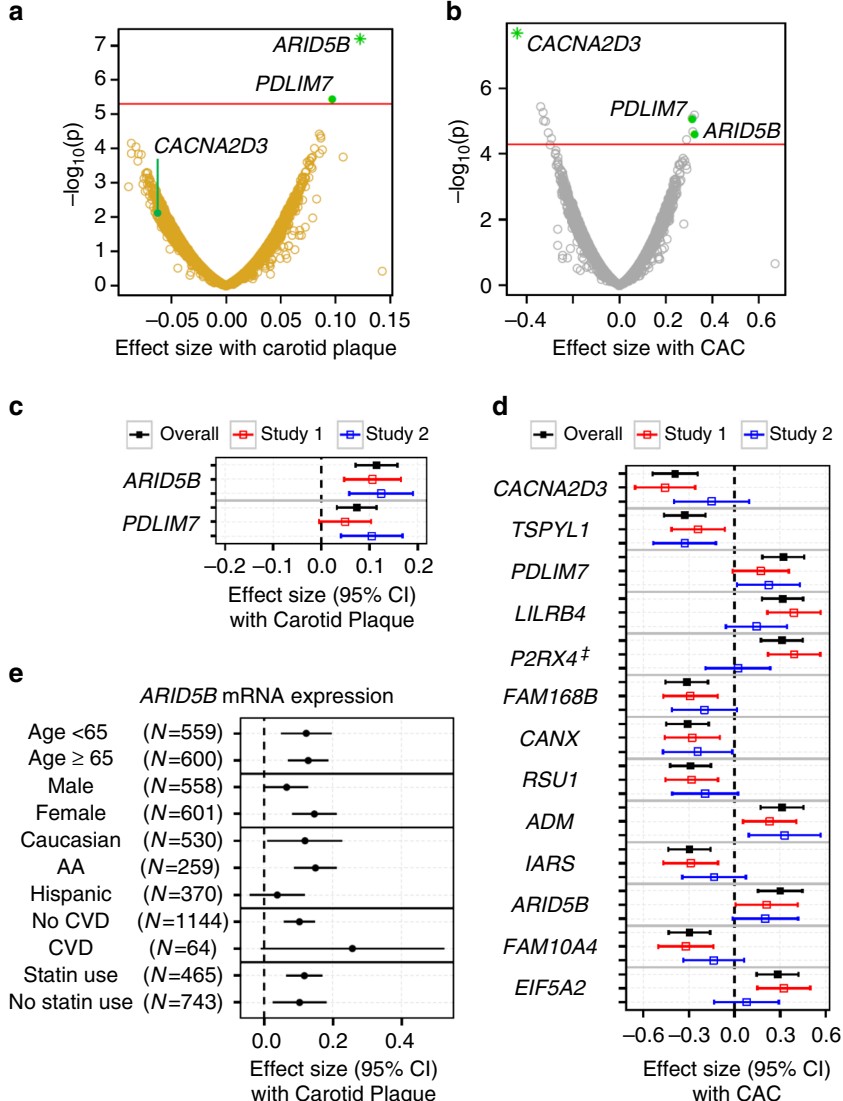

**Fig. 1** Transcriptomic associations with carotid plaque and CAC scores in 1208 MESA participants. The two volcano plots showing effect size ($\beta$) of $-\log_{10}(p\text{-value})$ for associations of each mRNA expression (for 10,989 unique mRNA expression) with (**a**) carotid plaque score and (**b**) CAC. The *red line* illustrates the threshold for an FDR $\leq 0.05$ (linear regression adjusting for age, sex, race, and study site). Expression of *ARID5B* and *PDLIM7* (*bolded*) were associated with both carotid plaque and CAC score; *green star* indicates mRNA expression most significantly associated with carotid plaque or CAC. The two forest plots showing the direction and effect size ($\beta$) for associations of each mRNA expression (in **a**, **b**) with (**c**) carotid plaque score and (**d**) CAC score in the full model (including traditional CVD risk factors), overall and by the two independent studies (study 1: JHU + CO, *red*; study 2: UMN + WFU, *blue*); ‡ significantly different effect sizes observed by study ($p_{interaction} < 0.05$). **e** The forest plot showing *ARID5B* mRNA expression associations with carotid plaque, stratified by age ($< 65$ years and $\geq 65$ years), sex, ethnicity/race, CVD, and statin use

contamination nor the % of CD14+ CD16+ cells was associated with the measures of atherosclerosis.

**Transcriptome signature of atherosclerosis**. Transcriptomic studies of atherosclerosis in mouse models and in humans[24–26] have been reported. However, no consensus atherosclerosis biomarkers or pathways have been identified. Of the 10,989 unique genes with RNA expression detectable in monocytes, we identified genes with expression associated with carotid plaque score ($n = 2$) and CAC ($n = 13$) at a $q$-value-based false discovery rate (FDR)[27] of 0.05 after adjusting for demographics (Fig. 1a, b). Using the FDR level of 0.05 for the genome-wide search, we would inevitably miss signals with small effect size due to the limited statistical power for the sample size we have; thus, additional signals for FDR of 0.20 were shown in Supplementary

Table 1 (21 gene transcripts with carotid plaque and 104 gene transcripts with CAC). Expression of two genes, *ARID5B* and *PDLIM7* (*PDZ and LIM domain protein 7*), were positively associated with both measures of atherosclerosis (FDR $\leq 0.05$); *ARID5B* was most significantly associated with carotid plaque score ($P = 6.30 \times 10^{-8}$, FDR $= 1.08 \times 10^{-3}$; with CAC: $P = 2.47 \times 10^{-5}$, FDR $= 0.03$). ARID5B has a key metabolic role in adipose, liver, and smooth muscle, and was previously implicated in lipid metabolism and adipogenesis in mice[28, 29]. PDLIM7 is an actin and protein kinase adaptor that promotes mineralization, which might be relevant to plaques with calcium. The most significant signal associated with CAC was lower expression of *CACNA2D3* (calcium channel, voltage-dependent, alpha 2/delta subunit 3; $P = 2.08 \times 10^{-8}$, FDR $= 3.19 \times 10^{-4}$; with carotid plaque: $P = 6.71 \times 10^{-3}$, FDR $= 0.40$) a tumor suppressor gene that can induce mitochondrial-mediated apoptosis[30]. In addition,

adjusting for other traditional CVD risk factors and statin use (in a model designated as the full model) had minimal impact on the significant associations (Fig. 1c, d). All of these associations had a consistent direction of effect across the independent study sites, in particular for three genes (*ARID5B*, *TSPYL1*, and *ADM*), which were cross-validated (*P* < 0.05, Fig. 1c, d). Race/ethnicity- and sex-specific analyses also were consistent across various strata (Supplementary Table 2). For the top signal, *ARID5B*, the

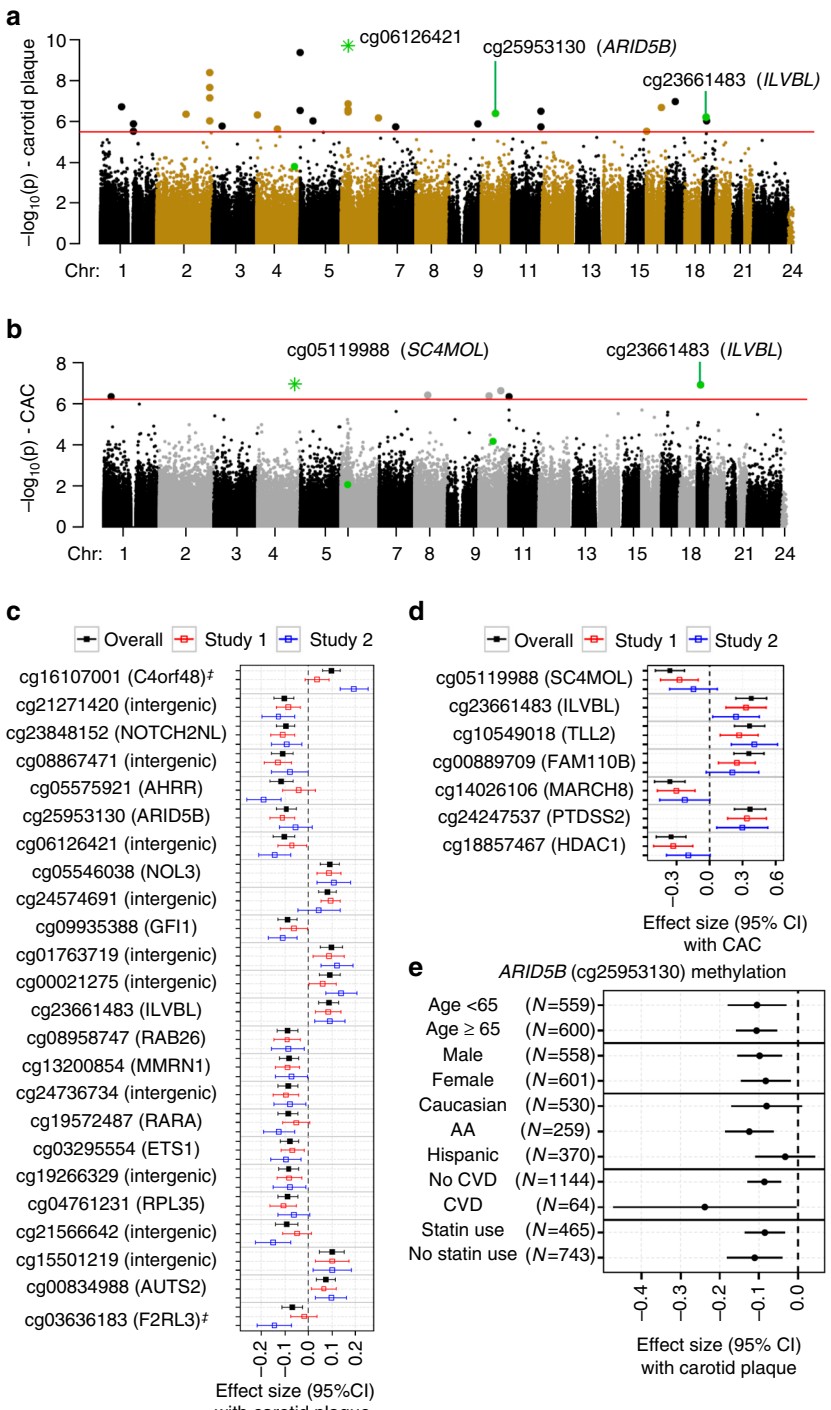

**Fig. 2** Methylomic associations with carotid plaque and CAC scores in 1208 MESA participants. The two Manhattan plots showing chromosomal locations of −log₁₀(*p*-value) for associations of each CpG site (for 484,817 CpG sites) with (**a**) carotid plaque score and (**b**) CAC score. The *red line* illustrates the threshold for an FDR ≤ 0.05; linear regression adjusting for age, sex, race, and study site). One CpG at *ILVBL* (*bolded*, *green dot*) was associated with both carotid plaque and CAC; *green star* indicates CpGs most significantly associated with carotid plaque or CAC. Notably, for *ARID5B*, both methylation (cg25953130, *green dot*) and mRNA expression were significantly associated with carotid plaque score. The two forest plots showing the direction and effect size (*β*) for associations of each CpG sites (in **a**, **b**, the most significant association at each unique loci is shown) with (**c**) carotid plaque score and (**d**) CAC in the full model (including traditional CVD risk factors), overall and by the two independent studies (study 1: JHU + CO, *red*; study 2: UMN + WFU, *blue*); ‡ significantly different effect sizes observed by study (*p*ₜₙₜₑᵣₐ𝒸ₜᵢₒₙ < 0.05). **e** The forest plot showing *ARID5B* cg25953130 methylation associations with carotid plaque score, stratified by age (< 65 years and ≥ 65 years), sex, ethnicity/race, CVD, and statin use

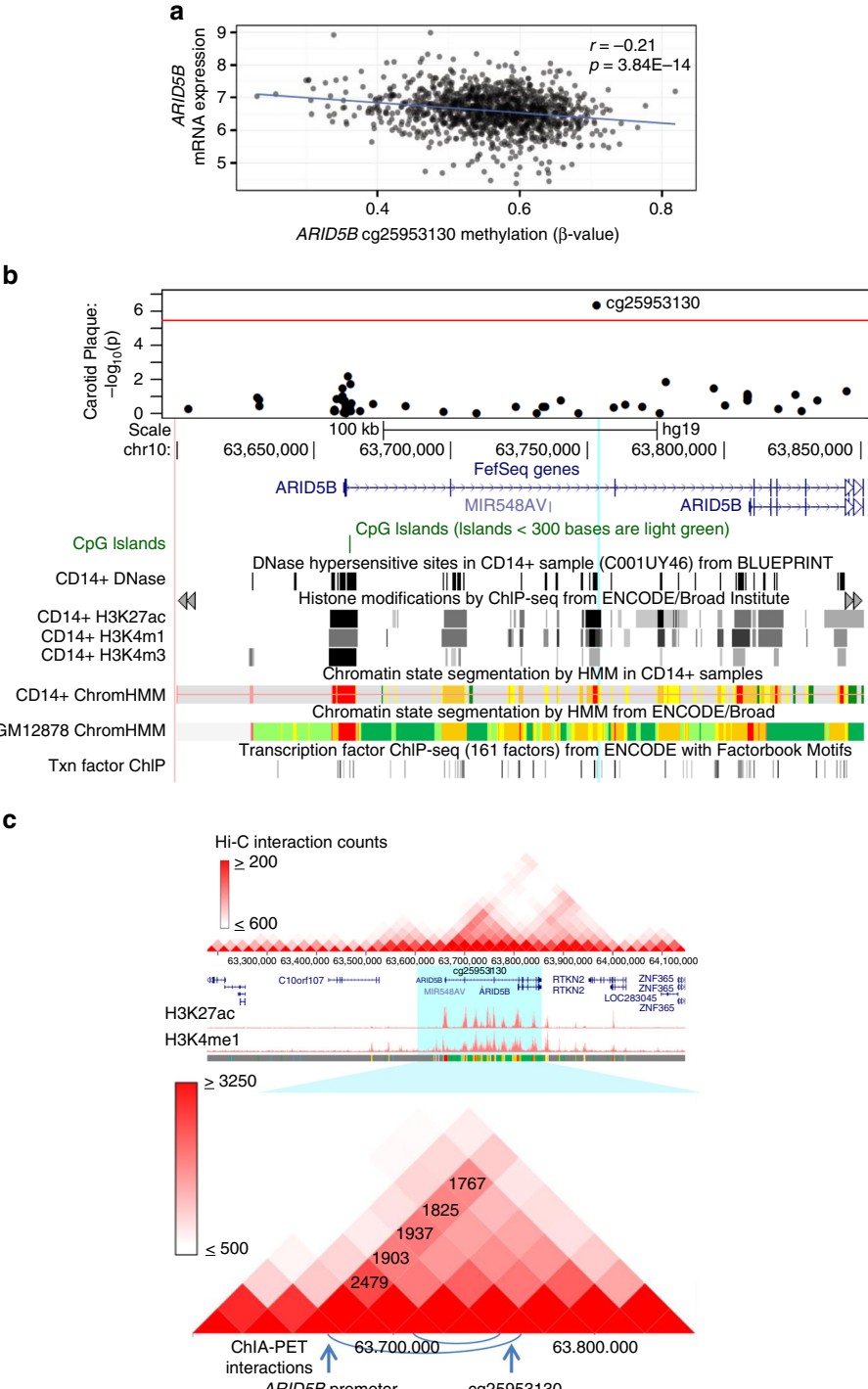

**Fig. 3** In vivo and in silico functional analysis of *ARID5B* CpG cg25953130. **a** *ARID5B* mRNA expression (*y* axis; normalized value) is significantly negatively correlated (Pearson's *r*) with methylation of a CpG (cg25953130) in 1264 CD14+ samples from MESA participants. **b** A regional association plot of *ARID5B* CpG methylation with carotid plaque score in MESA (*y* axis: −log$_{10}$ (*p*-value), *x* axis: position on chromosome (chr) 10) is shown in the *top panel*; the *bottom panel* shows the *ARID5B* expression-associated CpG (cg25953130, chr10:63,753,550, hg19, indicated by the *light blue line*) located in an *ARID5B* intron, overlaps a DNase hypersensitive site in a CD14+ sample from BLUEPRINT, histone marks indicative of a strong enhancer/promoter in a CD14+ and B-cell line sample from BLUEPRINT and ENCODE (see Supplementary Fig. 3 for ChromHMM color code), as well as a transcription factor binding site for EP300, detected in a neuroblastoma cell line (SK-N-SH_RA). **c** A physical interaction is detected between the *ARID5B* promoter and the region near the *ARID5B* CpG cg25953130 by both HiC and ChIA-PET data. The heatmap (*white*, low; *red*, high; *top panel*) graphically displays Hi-C interaction counts (the normalized number of contacts between a pair of loci) for the large region surrounding *ARID5B* in a B-cell line (GM12878; reported by Rao et al.[39]); active enhancer marks (H3K27ac and H3K4me1 in GM12878 from ENCODE) are shown below; the *bottom panel* zooms in on *ARID5B* (*blue highlighted region*) Hi-C interaction. There were 1,937 contacts between the *ARID5B* promoter and the cg25953130 region. Below the Hi-C interaction is a depiction of detected ChIA-PET interactions (chromatin interaction analysis by paired-end tag sequencing) for the region flanking *ARID5B* in the B-cell line (GM12878; reported by Heidari et al.[41]), represented by *blue curves*

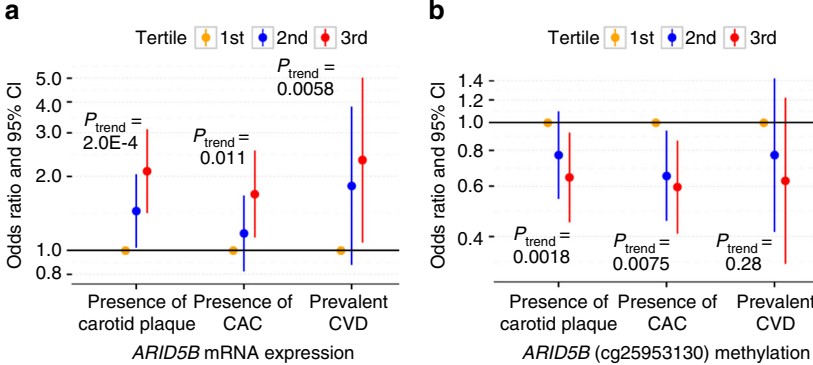

**Fig. 4** *ARID5B* expression and methylation are associated with subclinical and clinical CVD risk. Odds ratio of subclinical and clinical CVD (in the full model with adjustment of traditional CVD risk factors and statin use) by tertiles of **a** *ARID5B* mRNA expression and **b** *ARID5B* methylation (cg25953130)

associations with carotid plaque were consistent in direction and nominally significant across subgroups of age ( < or $\geq$ 65 years), sex, and statin use (Fig. 1e). Results utilizing RNA-sequencing-derived expression levels from a subset ($N = 354$) of the monocyte samples validated significant associations between *ARID5B* and CAC ($\beta \pm SE = 0.45 \pm 0.20$, $P = 0.03$) and carotid plaque ($\beta \pm SE = 0.13 \pm 0.06$, $P = 0.04$).

To identify network modules of highly correlated transcripts in an unbiased manner, we applied the weighted gene co-expression network analysis (WGCNA)[31] and identified 40 co-expressed gene network modules. Three modules significantly (FDR $\leq 0.05$) associated with CAC (Supplementary Table 3), including a cholesterol metabolism transcriptional network (CMTN) with 12 functionally coupled genes that was also associated with carotid plaque. The CMTN eigengene (the first principal component) of CMTN was associated with a transcriptional profile expected to increase intracellular cholesterol, including upregulation of cholesterol uptake (*LDLR* and *MYLIP*), and cholesterol and fatty acid synthesis genes (*HMGCS1*, *FDFT1*, *SQLE*, *CYP51A1*, *SC4MOL*, *SC5DL*, *SCD*, and *FADS1*), as well as downregulation of cholesterol efflux genes (*ABCG1* and *ABCA1*). We recently reported the CMTN being a signature feature of obesity, which was associated with CAC ($P = 3.34 \times 10^{-4}$)[13]. Here we further showed the CMTN was positively associated with carotid plaque ($P = 2.31 \times 10^{-4}$; FDR = 0.009). Among the CMTN members, *LDLR* expression was most significantly associated with CAC ($P = 1.34 \times 10^{-4}$, FDR = 0.10) and *ABCG1* expression was most significantly associated with carotid plaque ($P = 1.48 \times 10^{-4}$, FDR = 0.18, Supplementary Table 1). The other two network modules significantly (FDR $\leq 0.05$) associated with CAC were enriched with genes involved in phagosome formation (module 39; FDR = $1.32 \times 10^{-2}$) and migration of cells (module 23; FDR = $3.96 \times 10^{-2}$; Supplementary Table 3).

**Methylome signature of atherosclerosis.** Of the 484,817 CpG sites, 31 and 7 had methylation significantly associated with carotid plaque and CAC, respectively, including 1 CpG (cg23661483 in exon of *ILVBL*) associated with both carotid plaque and CAC (FDR $\leq 0.05$, adjusting for demographics, Fig. 2a, b and Supplementary Table 4 showing additional signals for FDR $\leq 0.10$). The most significant CpG (cg06126421) associated with carotid plaque score ($P = 2.00 \times 10^{-10}$, FDR = $8.91 \times 10^{-5}$) is an intergenic CpG, which recently was reported as one of the most significant smoking-associated methylation sites genome wide[32]. Notably, the carotid plaque-associated methylation sites include one CpG in *ARID5B* (cg25953130, intron 2, $P = 4.31 \times 10^{-7}$, FDR = 0.01), which tends to be hypomethylated in the individuals with higher carotid

plaque scores. Similar results were found between cg25953130 methylation and CAC ($P = 6.80 \times 10^{-5}$, FDR = 0.32).

The 37 differentially methylated sites associated with carotid plaque or CAC presented higher variable CpG methylation levels (across MESA population) with the interdecile range of the percentage of methylation (measured by $\beta$-value[33]) ranging from 4 to 24% (median: 10%), compared with the whole methylome (median: 4%). The majority of the methylation sites associated with carotid plaque or CAC had inverse associations with the atherosclerosis measures (Supplementary Table 4). The 37 atherosclerosis-associated CpGs are distributed among 30 unique genomic loci.

Additional adjustments in the full model had minimal impact on the significant associations, which were also consistent across the two study sites for the majority (61%) of differentially methylated sites ($P < 0.05$, as shown for each unique loci in Fig. 2c, d). Race/ethnicity- and sex-specific analyses also showed high consistency across the various strata (Supplementary Table 5), suggesting shared effects across ancestries. At the genome-wide level, race/ethnicity-specific analyses did not uncover additional significant associations. The associations of *ARID5B* methylation (cg25953130, intron) with carotid plaque were consistent in direction and nominally significant across subgroups of age ( < or $\geq$ 65 years), sex, CVD status, and statin use (Fig. 2e). Pyrosequencing derived methylation of cg25953130 from a subset ($n = 90$) of the monocyte samples significantly correlated with microarray-based methylation levels ($r = 0.92$, $P = 5.2 \times 10^{-37}$) and validated significant associations between cg25953130 methylation and carotid plaque ($\beta \pm SE = -0.07 \pm 0.03$, $P = 6.31 \times 10^{-3}$).

**In vivo and in silico functional validation.** DNA methylation has been viewed as an important potential regulator of gene expression[34]. To prioritize the list of differentially methylated CpGs, we assessed expression-associated methylation sites (eMS) reported in the same monocyte samples by our previous study[22]. We identified five atherosclerosis-associated CpGs whose methylation was significantly related to messenger RNA expression of at least one nearby gene (Supplementary Table 6). Out of the five atherosclerosis-associated eMS, four had methylation correlated with mRNA expression profiles that were nominally ($P < 0.05$) associated with atherosclerosis, including expression of *SC4MOL*. The CpG most significantly associated with CAC (cg05119988, located in the 5'-untranslated region of *SC4MOL*) significantly correlated ($r = -0.17$, $P = 1.2 \times 10^{-10}$, FDR = $7.3 \times 10^{-8}$) with mRNA expression of *SC4MOL*, a member of the identified CMNT, which was nominally associated with both carotid plaque and CAC. The *SC4MOL* eMS resides in a predicted

weak promoter region of a B-cell line; however, this region was identified as heterochromatin in monocytes (using ChromHMM[35] for histone modification data from BLUEPRINT[36, 37], Supplementary Fig. 3 and Supplementary Tables 4 and 6).

The *ARID5B* CpG (cg25953130) was the only CpG site significantly associated with expression of *ARID5B* ($P = 3.84 \times 10^{-14}$, FDR = $5.70 \times 10^{-6}$; Fig. 3a). As shown in Fig. 3b, it overlaps a DNase hypersensitive hotspot (BLUEPRINT

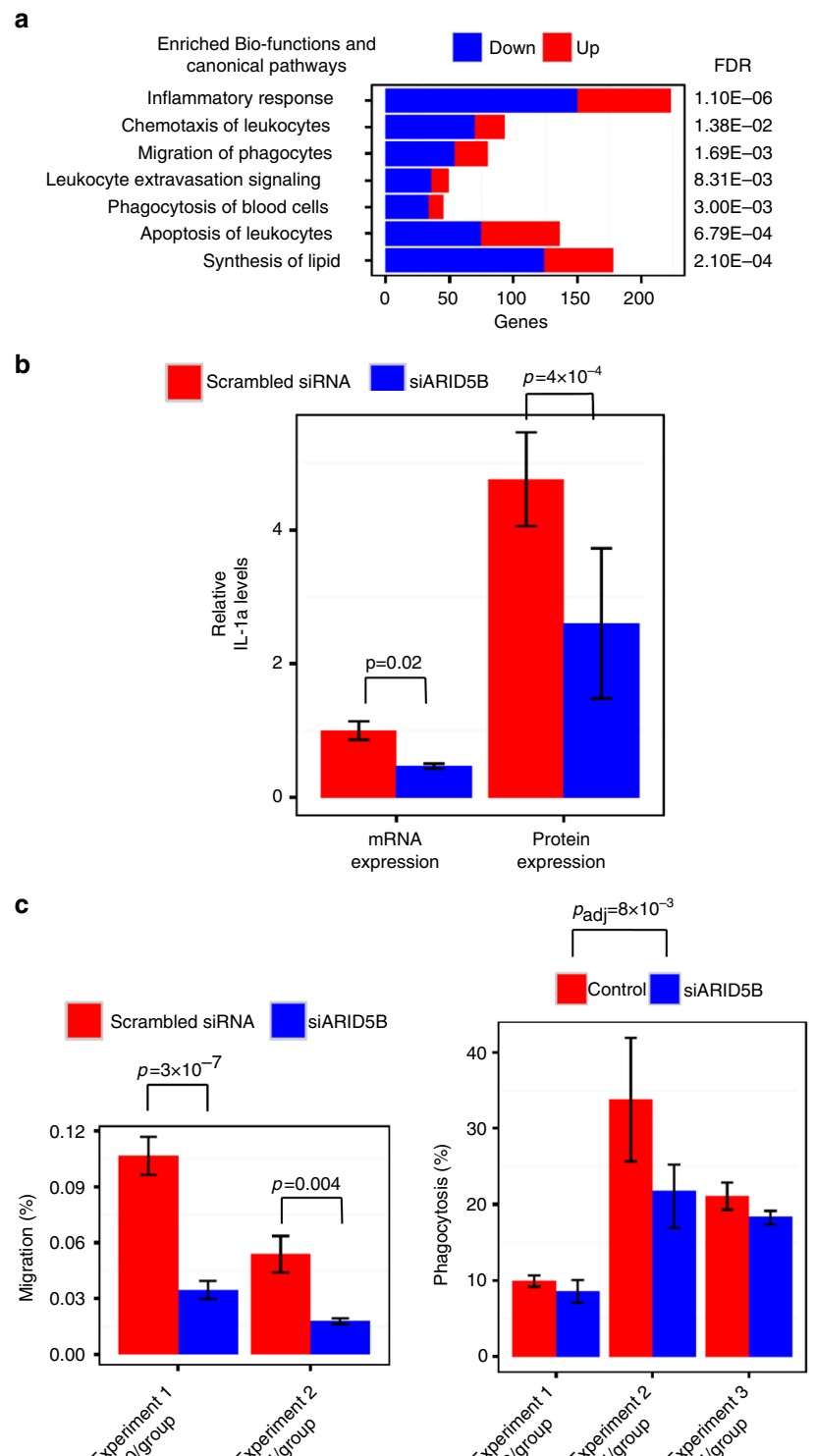

**Fig. 5** siRNA knockdown of *ARID5B* alters immune/inflammatory response and lipid metabolism genes. After 3 h of LPS treatment (100 ng ml$^{-1}$) of THP1-monocyte *ARID5B* knockdown samples compared with control samples (scrambled siRNA), **a** the transcriptome ($N = 8$ per group) were enriched with the inflammatory response genes including the listed bio-functions and canonical pathways (enrichment FDR < 0.05 from IPA); proportion of genes upregulated shown as red, downregulated shown as *blue*. **b** Relative IL-1A levels (mean ± SD) of mRNA expression in cells ($n = 3$ per group, by RT-PCR) and protein expression in culture media ($n = 3$ per group, by ELISA) decreased. **c** Cell migration (mean ± SD, $n = 10$ and 4 per group in experiment 1 and 2, respectively), and phagocytosis (mean ± SD, $n = 8$, 4, and 4 per group in experiment 1, 2, and 3, respectively) were inhibited

monocyte data[36, 37]), a predicted strong enhancer (using both monocyte and B-cell line histone mark data from the BLUEPRINT[36, 37] and ENCODE[38] projects, respectively), and a transcription factor-binding site occupied by EP300 (in a neuroblastoma cell line). More importantly, chromatin-capture sequencing technologies (both Hi-C and ChIA-PET) confirmed direct interactions between regions in the *ARID5B* cg25953130 locus and the *ARID5B* promoter region in B-cell line[39–42] (Fig. 3c). Our data, together with the publically available functional data, strongly support the presence of an *ARID5B* regulatory region in the *ARID5B* gene body flanking *ARID5B* cg25953130.

To test whether the assumed methylation effects on atherosclerosis burden were mediated through its associated mRNA expression, we used structural equation modelling (SEM) with bootstrapping (R package *lavaan*[43]) to perform mediation analyses. We showed that *ARID5B* mRNA expression significantly mediated 15% and 14% of the total effect of this *ARID5B* CpG on carotid plaque score (indirect effect, $P = 2.1 \times 10^{-4}$) and CAC ($P = 2.1 \times 10^{-3}$), respectively.

The inverse association between methylation of this *ARID5B* CpG and carotid plaque score after adjusting for *ARID5B* expression (direct effect) was also significant. Jointly, the associations of *ARID5B* gene expression and methylation levels with atherosclerosis explain an additional 2.3% of the variability in carotid plaque score beyond well-known CVD risk factors and statin use. The effect sizes of *ARID5B* gene expression and methylation levels associating with carotid plaque score are higher than the effect sizes of T2DM (1% of variability) or hypertension (0.9%) in the same model. These data suggest that different types of related genomic features (mRNA expression and DNA methylation) may offer additive values in prediction of CVD susceptibility.

To further demonstrate the clinical relevance of *ARID5B*, we associated the tertiles of *ARID5B* mRNA expression and *ARID5B* cg25953130 methylation with presence of carotid plaque (defined as carotid plaque score greater than zero, N = 816 cases), presence of CAC (defined as CAC > 0, N = 844 cases), and prevalent CVD (history of a coronary heart event or stroke, N = 64 cases). *ARID5B* mRNA expression was positively associated with the presence of carotid plaque (third tertile odds ratio = 2.10, 95% confidence interval (CI): 1.42–3.09, $P = 1.87 \times 10^{-4}$), presence of CAC (third tertile odds ratio = 2.10, 95% CI: 1.42–3.09, $P = 1.87 \times 10^{-4}$), and prevalent CVD (third tertile odds ratio = 2.33, 95% CI: 1.08–5.02, $P = 3.12 \times 10^{-2}$, Fig. 4a), whereas *ARID5B* cg25953130 methylation was inversely associated with the presence of carotid plaque (third tertile odds ratio = 0.64, 95% CI: 0.45–0.92, $P = 1.59 \times 10^{-2}$) and presence of CAC (third tertile odds ratio = 0.64, 95% CI: 0.45–0.92, $P = 1.59 \times 10^{-2}$, Fig. 4b) in the full model. To examine the dose–response relationship between *ARID5B* and extent of atherosclerosis, which may indicate their potential contribution to the progression of plaques, we performed linear regression analysis while excluding those with zero value of carotid plaque score. The associations with carotid plaque score remain significant for the *ARID5B* mRNA expression (unique variability explained: 1.2%, $P = 1.1 \times 10^{-3}$) and cg25953130 methylation (unique variability explained: 0.73%, $P = 1.1 \times 10^{-2}$).

**Identified genomic features and known CVD risk factors**. Among the atherosclerosis-associated genomic features, eight mRNA (Supplementary Table 7) and 33 CpGs (Supplementary Table 8) were also associated with one or more traditional CVD risk factors, particularly demographics (age, sex, and race/ethnicity), cigarette smoking, and obesity. For three mRNA and 29 CpGs (*bolded* in Supplementary Table 7 and Supplementary

Table 8), the predicted effects of the majority of CVD risk factors on atherosclerosis that were mediated through the genomic features have the similar direction as the observed associations between the CVD risk factors and the measure of atherosclerosis.

*ARID5B* expression was positively associated with many CVD risk factors (FDR < 0.05 with adjustment for demographic variables), such as age, ($P = 2.06 \times 10^{-13}$), BMI ($P = 1.67 \times 10^{-8}$), T2DM ($P = 3.43 \times 10^{-5}$), and inflammatory stress (measured by plasma interleukin-6 levels (IL-6; $P = 1.36 \times 10^{-10}$), and inversely associated with HDL-C levels ($P = 6.86 \times 10^{-6}$). *ARID5B* methylation (cg25953130) was associated inversely with age ($P = 3.33 \times 10^{-11}$) and plasma IL-6 levels ($P = 0.004$), and the *ARID5B* CpG tended to be hypomethylated in current smokers ($P = 7.19 \times 10^{-7}$). Although we cannot be certain of causality, if we assume *ARID5B* to be in fact causal, mediation analyses showed the expression of *ARID5B* significantly mediated 10 and 25% of the total effect of age and IL-6 on carotid plaque score ($P = 7.46 \times 10^{-6}$, $4.33 \times 10^{-5}$, respectively), and the *ARID5B* methylation significantly mediated 7 and 10% of the total effect of age and IL-6 on carotid plaque score ($P = 2.57 \times 10^{-4}$, $8.28 \times 10^{-3}$, respectively).

***ARID5B* RNA expression and methylation in CD4+ T cells**. To examine the *ARID5B* expression and methylation across cell types, similar analyses were performed in a subset of 517 MESA CD4+ T-cell samples. Correlations for *ARID5B* expression and methylation between monocyte and T cells were weak ($r = 0.12$ and 0.30, respectively). Within T-cell samples, *ARID5B* expression was inversely correlated with cg25953130 methylation ($r = -0.45$, $P = 1.27 \times 10^{-31}$), as seen in monocytes. The *ARID5B* mRNA and cg25953130 methylation associations with carotid plaque and CAC were not significant in T-cell samples ($P > 0.05$), but remained statistically significant ($p$ ranges from $3.8 \times 10^{-3}$ to $7.2 \times 10^{-3}$) when analyzed in the monocyte samples from the same subset. In the same data, however, we observed the association of *AHRR* cg05575921 with carotid plaque score and CAC score in T-cell samples ($P = 3.18 \times 10^{-5}$ and $5.89 \times 10^{-5}$, respectively), with a strong correlation between the two cell types for the *AHRR* methylation site ($r = 0.97$). *AHRR* hypomethylation is a well-known, robust biomarker of smoking that we recently linked to carotid plaque score in MESA[14]. These results demonstrate examples of both potential cell-type-specific and shared genomic features in relation to burden of atherosclerosis.

**Functional evaluation of *ARID5B* using in vitro models**. Although little is known about the majority of identified genomic features, pleotropic effects of ARID5B in adipogenesis[28], chondrogenesis[44], autoimmune diseases[45, 46], lipid metabolism[29], and smooth muscle cell differentiation[47] have been previously reported. As a transcription coactivator that is part of the H3K9me2 demethylase complex with PHD finger protein 2, ARID5B is expected to activate its target genes by removing the repressive H3K9ME2 histone mark (demethylation of H3K9me2) from the promoter region of target genes[44, 48]. To evaluate the functional role of ARID5B in monocytes/macrophages, we examined the effects of its knockdown on transcriptomic profiles in lipopolysaccharide (LPS)-stimulated human THP1 monocytes in two independent experiments (LPS: 100 ng ml$^{-1}$, $n = 4$ per group at each experiment).

*ARID5B* mRNA knockdown efficiency was 85 and 76% in the two experiments (Supplementary Fig. 4); both consistently showed that *ARID5B* knockdown decreased expression of 1,320 genes and increased expression of 1,162 genes (FDR < 0.005 in the discovery set and FDR < 0.05 in the replication set; Supplementary Data 1).The 2,482 ARID5B-modified genes

displayed a significant overrepresentation of related pathways including inflammatory/immune response, chemotaxis, migration, extravasation signaling, and phagocytosis, and also the lipid synthesis functional pathway, compared with the background list of genes detectable on the array (enrichment FDR < 0.05, Fig. 5a, full list of genes provided in Supplementary Table 9). Enriched biofunctions and canonical pathways in the inflammatory response pathway consisted of mainly downregulated genes, including key proinflammatory cytokines (e.g., tumor necrosis factor (TNF) and IL-1a), activator and effector cytokines from the type I interferon signaling pathway (e.g., IRF3, IFNB1, STAT1, and STAT2), and antigen processing and presentation genes (e.g., major histocompatibility complex class II cell surface receptors: HLA-DRA and HLA-DRBs, Supplementary Fig. 4 and *bolded* in Supplementary Table 9), collectively indicating decreased pathway activation. Decreased expression of genes in two of the enriched pathways, phagocytosis, and lipid synthesis (shown in red in Supplementary Table 9), overlapped with the members of two atherosclerosis-associated networks we identified in MESA (CMTN and module 39—enriched for the phagosome formation pathway, Supplementary Table 3), which suggests ARID5B may contribute to the gene network associations with atherosclerosis.

To interrogate the potential pro-inflammatory role that ARID5B may have, we repeated the third ARID5B knockdown experiment and performed enzyme-linked immunosorbent assay (ELISA) assays of culture media for IL-1A, the pro-inflammatory cytokine that was most significantly reduced by ARID5B mRNA knockdown (FDR: $3.79 \times 10^{-8}$, $4.48 \times 10^{-4}$ in the first two experiments, Supplementary Data 1). ARID5B knockdown decreased levels of both IL1A mRNA ($P = 0.02$) in the THP1-monocytes and IL-1A protein expression ($P = 4.28 \times 10^{-4}$) in their culture media, as shown in Fig. 5b.

To further examine the effects of ARID5B knockdown on cellular functions suggested by the transcriptomic profile changes, we performed THP1-monotye migration and phagocytosis assays. ARID5B knockdown suppressed monocyte migration ($P = 3 \times 10^{-7}$, 0.004 in experiment 1 and 2, respectively, Fig. 5c). ARID5B knockdown also moderately inhibited monocyte phagocytosis ($P = 0.008$) as shown in Fig. 5c.

## Discussion

In summary, using purified blood monocytes (CD14+) we discovered many epigenome and transcriptome features of atherosclerosis independent of well-known CVD risk factors, including several robust signals that were cross-validated by two state-of-art imaging measures of atherosclerosis, at different study sites, in two different arterial beds, and in both males and females. We further demonstrated that most of the atherosclerosis-associated genomic features were associated with traditional CVD risk factors, but not reported genetic risk variants. Although causal directions could not be inferred in our observational study, we have prioritized genomic features that link to well-established CVD risk factors and atherosclerosis, implicating the underlying mechanisms of traditional CVD risk factors (e.g., age and obesity) in atherogenesis partially mediated through ARID5B. Our data suggest two pathways for further investigation: (1) whether in the causal pathway or reflective of disease, epigenome, and transcriptome features in circulating monocytes are potential "biosensors" that might be useful for detecting early signs of metabolic disorders and elevated CVD risk; and (2) if in the causal pathway, several new candidates are emerging for potential intervention. Our study contributes to a larger list of atherosclerosis biomarkers and provides accessibility to unique multi-omics databases with well-characterized clinical phenotypes.

Finally, we contribute to the rapidly evolving picture of human gene expression regulation that promises to be formidable in both scope and complexity[49].

In particular, we identified a novel molecular link between ARID5B (transcription coactivator for histone H3K9me2 demethylation) mRNA expression and atherosclerosis, as well as the ARID5B CpG DNA methylation alteration that was inversely associated with both ARID5B expression and atherosclerosis. It is worth noting that the observed inverse association of the ARID5B methylation site with its expression is not sufficient to support its causal role, as methylation change could be caused by additional transcription factors binding to this enhancer region. Local fine-mapping of the methylation patterns for ARID5B cg25953130 loci is also needed. Taken together with the existing chromatin-capture sequencing and histone marks data, the integration of our methylomic and transcriptomic data identifies the intronic region, where the ARID5B CpG (cg25953130) resides, most likely being in the ARID5B enhancer. Our mediation analysis further suggests that the ARID5B methylation association with atherosclerosis is possibly via an epigenetically controlled enhancer that supports promoter activation and ARID5B transcription.

Importantly, our in vitro experiments support a novel functional role of ARID5B in immune and metabolic homeostasis. We found that ARID5B knockdown limited LPS-stimulated gene expression increases in pro-inflammatory and interferon signaling pathways[50]. The ARID5B-regulated genes are expected to increase the activation state of atherosclerosis-relevant functions, including leukocyte chemotaxis, migration, extravasation, and phagocytosis, as well as lipid synthesis. We further demonstrated ARID5B knockdown inhibited monocyte migration and phagocytosis when deregulated by persistent stress signals. Chronic inflammation and its related functional changes in monocytes are well-recognized mechanisms in atherosclerosis[9, 10]. The ARID5B-dependent pro-inflammatory response was consistent with the positive correlation between ARID5B expression and plasma IL-6 levels that we observed in MESA. We also confirmed the role of ARID5B in lipid metabolism that implicated previously[29]. Collectively, our findings suggest that increased ARID5B expression promotes atherosclerosis by dysregulating immunometabolism towards a chronic inflammatory phenotype.

Given the role of ARID5B in chromatin state dynamics[44, 48], we speculate that ARID5B with its histone cofactor PHF2 demethylase are at a pivotal gene regulatory axis that epigenetically controls immune and metabolic homeostasis. Conceptually, altering the ARID5B-regulated H3K9me2 repressor state known to exist for proinflammatory gene silencing in monocytes[51] may convert poised enhancers and promoters to an active state[52], therefore directing expression of the gene pathways observed in this population and the in vitro study. Searching for predicted ARID5B binding site motifs (conserved in the human, mouse, and rat sequence alignments), we identified potential ARID5B-binding sites within 20 kb of 760 genes with decreased expression following ARID5B knockdown, which include key genes for proinflammation (e.g., TNF, IL-1α, and IFNB1). Chronic inflammation perturbs immunometabolic homeostasis, contributing to the development of many common diseases and to the aging process. Further investigation of the role of the ARID5B-PHF2 epigenetic axis in chronic inflammation is critically needed. Finally, our findings indicate that blood monocytes obtained from large epidemiologic investigations of normal vs. chronic inflammation states such as atherosclerosis can be combined with human cell culture models to illuminate molecular immunometabolic pathways and inform novel targets for prevention and intervention of atherosclerosis.

## Methods

**Participants.** The present analyses are primarily based on data collected at MESA Exam 5 (April 2010–February 2012) with concurrent analyses of purified monocyte samples of 1264 randomly selected MESA participants from four MESA sites (John Hopkins University, Columbia University, University of Minnesota, and Wake Forest University). The study protocol was approved by Institutional Review Boards of the four institutions. All participants signed informed consent.

**Measurement of atherosclerosis and prevalent CVD.** All measurements in humans were obtained at MESA Exam 5 unless otherwise specified. To obtain carotid artery plaque scores, readers at the UW Atherosclerosis Imaging Research Program Laboratory adjudicated carotid plaque presence or absence, defined as a focal abnormal wall thickness (carotid IMT > 1.5 mm) or a focal thickening of > 50% of the surrounding IMT, as reported previously[19, 53]. The presence or absence of plaque acoustic shadowing was recorded. A total plaque score (range 0–12) was calculated to describe carotid plaque burden. One point per plaque was allocated for the near and far walls of each segment (common carotid artery, bulb, and internal carotid artery) of each carotid artery that was interrogated. For carotid plaque presence, intra-reader reproducibility was excellent (per reader $\kappa = 0.82$–1.0, overall $\kappa = 0.83$, 95% CI 0.70–0.96), as was inter-reader reproducibility ($\kappa = 0.89$; 95% CI 0.72–1.00)[54]. The CT Reading Center for cardiac scans in the MESA is at UCLA-Biomedical Research Institute. Coronary artery calcium (CAC) score was determined using the Agatston method, which accounts for both lesion area and calcium density using Hounsfield brightness. The re-read agreement for the CAC score (intraclass correlation coefficient = 0.99) was excellent. Prevalent CVD was defined as a past history of myocardial infarction, angina (which included definite angina and probable angina if coronary revascularization was performed at the same time or afterwards), resuscitated cardiac arrest, or stroke.

**Measurement of covariates.** Weight was measured with a Detecto Platform Balance Scale to the nearest 0.5 kg. Height was measured with a stadiometer (Accu-Hite Measure Device with level bubble) to the nearest 0.1 cm. BMI was defined as weight in kilograms divided by square of height in meters. T2DM was defined as fasting glucose $\geq 7.0$ mmol l$^{-1}$ ($\geq 126$ mg dl$^{-1}$) or use of hypoglycemic medication, and impaired fasting glucose was defined as fasting glucose 5.6–6.9 mmol l$^{-1}$ (100–125 mg dl$^{-1}$). Plasma IL-6 was measured by ultra-sensitive ELISA (Quantikine HS Human IL-6 Immunoassay; R&D Systems, Minneapolis, MN). Plasma C-reactive protein (CRP) was measured using the BNII nephelometer (High-Sensitivity CRP; Dade Behring Inc., Deerfield, IL). Resting blood pressure was measured three times in the seated position using a Dinamap model Pro 100 automated oscillometric sphygmomanometer (Critikon, Tampa, FL). The average of the last two measurements was used in analysis. Hypertension was defined as systolic pressure $\geq 140$ mm Hg, diastolic pressure $\geq 90$ mm Hg, or current use of anti-hypertensive medication. The single nucleotide polymorphisms (SNPs) data were derived from MESA Affymetrix 6.0 array genotype data[55].

**Blood cell count and purification of CD14+ and CD4+ cells.** Samples for complete blood count with differential analysis were obtained by venipuncture and collected into tubes containing EDTA. Total circulating WBC count and cell subtype counts were performed at local LabCorp. Blood was also collected in sodium heparin-containing Vacutainer CPT cell separation tubes (Becton Dickinson, Rutherford, NJ) to separate peripheral blood mononuclear cells from other elements within two hours from blood draw. Subsequently, monocytes and T cells were isolated with anti-CD14 and anti-CD4 monoclonal antibody-coated magnetic beads, respectively, using an autoMACS automated magnetic separation unit (Miltenyi Biotec, Bergisch Gladbach, Germany). Initially, flow cytometry analysis of 18 specimens was performed, including samples from all four MESA field centers, which were found to be consistently > 90% pure.

**DNA/RNA extraction.** DNA and RNA were isolated from samples simultaneously using the AllPrep DNA/RNA Mini Kit (Qiagen, Inc., Hilden, Germany). DNA and RNA quality control (QC) metrics included optical density measurements, using a NanoDrop spectrophotometer and evaluation of the integrity of 18 and 28 s ribosomal RNA using the Agilent 2100 Bioanalyzer with RNA 6000 Nano chips (Agilent Technology, Inc., Santa Clara, CA) following manufacturer's instructions. RNA with RIN (RNA Integrity) scores > 9.0 was used for global expression microarrays. The median of RIN for our 1264 samples was 9.9.

**Global mRNA expression quantification.** The Illumina HumanHT-12 v4 Expression BeadChip and Illumina Bead Array Reader were used to perform the genome-wide expression analysis, as previously described[22]. This data has been deposited in the NCBI Gene Expression Omnibus and is accessible through GEO Series accession number (GSE56047).

**Epigenome-wide methylation quantification.** Illumina HumanMethylation450 BeadChips and HiScan reader were used to perform the epigenome-wide methylation analysis, as previously described[22]. This methylation data has been deposited in the NCBI Gene Expression Omnibus and is accessible through GEO Series accession number (GSE56046).

**QC and pre-processing of microarray data.** Data pre-processing and QC analyses were performed in R (http://www.r-project.org/) using Bioconductor (http://www.bioconductor.org/) packages, as previously described[22]. For both monocyte and T-cell assays, we included 2% blind duplicates. Correlations among technical replicates exceeded 0.997. Multidimensional scaling plots showed the five common control samples were highly clustered together and identified three outlier samples, which were excluded subsequently.

The Illumina HumanHT-12 v4 Expression BeadChip included > 47,000 probes for > 30,000 genes (with unique Entrez gene IDs). Statistical analyses excluded probes with non-detectable expression in $\geq 90\%$ of MESA samples (using a detection $p$-value cutoff of 0.0001), probes overlapping repetitive elements or regions, probes with low variance across the samples (< 10th percentile), or probes targeting putative and/or not well-characterized genes, i.e., gene names starting with KIAA, FLJ, HS, MGC, or LOC; 14,619 mRNA transcripts from 10,989 unique genes were included in the analyses of the presented manuscript.

The Illumina HumanMethylation450 BeadChip included probes for 485 K CpG sites. Exclusion criteria included probes with "detected" methylation levels in < 90% of MESA samples using a detection $p$-value cutoff of 0.05 and 65 control probes which assay highly polymorphic SNPs rather than DNA methylation[56]. Methylation data for the total of 484,817 CpG sites were included in the analyses.

To estimate residual sample contamination, we generated separate enrichment scores for neutrophils, B cells, T cells, and natural killer cells as described previously[22]. We adjusted for residual sample contamination with non-monocyte cell types in all the analyses. Although most of monocytes (80–90%) are expected to be CD14+ CD16–[23], we also adjusted for expression of the *FCGR3a* gene (*CD16a*).

**RNA sequencing.** A subset of 374 samples was randomly selected from the 1,264 MESA monocyte samples for RNA sequencing (see the Supplementary Methods for details).

**Bisulfite treatment of genomic DNA and pyrosequencing.** A subset of 90 samples was selected from the 1264 MESA monocyte samples, based on carotid plaque score extremes, matched for age, sex, and race (see the Supplementary Methods for details).

**Weighted gene co-expression network analyses.** The WGCNA method was used to construct network modules of highly correlated transcripts, using the R package WGCNA[31]. A total of 1261 MESA samples were included in the co-expression network analysis after removing 3 outlier samples based on hierarchical clustering and 13,196 mRNA transcripts were included. First, we constructed an unsigned weighted network based on the pairwise correlations among all transcripts considered, using a soft thresholding power of 5, chosen to produce approximately a scale-free topology. Then, using the topological overlap measure to estimate the network interconnectedness, the transcripts were hierarchically clustered. We used the default parameters of WGCNA, except for changing the correlation type from Pearson to biweight midcorrelation (which is more robust to outliers), the deepSplit setting from 2 to 3, the detectCutHeight value from 0.995 to 0.999, the maximum block size from 5000 to 14,000 transcripts, and the minimum size for module detection from 20 to 10.

WGCNA produces a set of modules, each containing a unique set of mRNA transcripts. The module eigengene was obtained to represent each module, which corresponds to the first eigenvector of the within-module expression correlation matrix (or the first right-singular vector of the standardized within-module expression matrix).

**Association analyses.** The overall goal of the association analysis was to characterize the associations of each measure of atherosclerosis (carotid plaque and CAC) with each of genome-wide measures of mRNA expression and DNA methylation. Analyses were performed using the linear model (*lm*) function of the *Stats* package and the *stepAIC* function of the *MASS* package in R. Reported correlations ($r$) represent the partial Pearson product-moment correlation coefficient. We fit separate linear regression models for each measure of atherosclerosis, with (1) genome-wide (log$_2$ transformed) mRNA expression profiles, (2) network module eigengenes from WGCNA, and (3) genome-wide DNA methylation profiles (*M*-values). The *M*-value is well suited for high-level analyses and can be transformed into the $\beta$-value, an estimate of the percent methylation of an individual CpG site that ranges from 0 to 1 ($M$ is logit($\beta$-value)). The $p$-values were adjusted for multiple testing using the $q$-value FDR method[27] and Benjamini–Hochberg FDR[57] when applicable. All analyses accounted for effects of residual cell contamination and covariates including age, gender, ethnicity, and study site. The full model also adjusted for traditional CVD risk factors (cigarette smoking, BMI, HDL-C and LDL-C levels, hypertension, and diabetes mellitus) and statin use.

**Mediation analysis.** To investigate the genomic features as a potential molecular link between CVD risk factors and extent of atherosclerosis, we performed mediation analyses under an assumed causal model in which a CVD risk factor leads to a change in the genomic feature, which at least partially mediates the effects of the CVD risk factor on atherosclerotic burden. The mediation analyses

accounted for the biological and technical covariates and were performed by robust SEM as implemented in the R package lavaan[43]. SEM analysis in general, and as implemented in lavaan, is based on maximum likelihood and the normal distribution, but provides several approaches to effectively deal with non-normal data. A first approach consists of computing robust SE by sandwich-type covariance matrices and scaled test statistics, in particular the Satorra–Bentler statistic whose amount of rescaling reflects the degree of kurtosis, whereas the second approach uses specific bootstrapping methods to obtain both SE and test statistics[43]. The results we report are based on bootstrapping which we found to be somewhat more conservative than the use of robust SE and the Satorra–Bentler statistic.

**In vivo functional annotation analysis**. CpGs with methylation significantly associated with carotid plaque scores or CAC were investigated for association with cis-gene expression by performing a look-up in the results from our previous analysis[22] of the same samples. Briefly, to identify DNA methylation associated with gene expression in cis, we fit separate linear regression models with the M-value for each CpG site (adjusted for methylation chip and position effects) as a predictor of transcript expression for any autosomal gene within 1 Mb of the CpG in question. Covariates were age, sex, and race/ethnicity, and study site.

mRNA expression and DNA methylation of the most significantly associated mRNA and CpGs associated with carotid plaques scores and CAC were also investigated for association with nearby genetic variants. We arbitrarily chose to investigate a large window ($\pm 1$ Mb) surrounding the mRNA/CpG to avoid missing any potential effects. Covariates were age, sex, and race/ethnicity, and study site. We fit separate linear regression models with SNPs located within 1 Mb as a predictor of the mRNA expression ($\log_2$ transformed) or the methylation (M-value) in the MESA samples from Caucasian participants, including SNPs with a minor allele frequency > 0.05 in the MESA Caucasian population. P-values were adjusted for multiple testing using the q-value FDR method[27].

**In silico functional annotation analysis**. For the differentially expressed co-expression modules, Ingenuity Pathway Analysis (IPA, QIAGEN, Redwood City, www.qiagen.com/ingenuity) was used to examine the enrichment of canonical pathways and biofunctions. In silico functional prediction of chromatin states in monocytes was performed using ChromHMM[35] to predict segmentation among six states, based on histone modifications in monocyte samples from the BLUEPRINTproject[36, 37] (H3K27ac, H3K4me1, and H3K4me3) and ENCODE[38] (H3K36me3). Other functional information utilized includes DNase hypersensitive hotspot data in a monocyte sample (C001UY46) from the BLUEPRINT project[36, 37], and transcription factor binding sites detected in any cell type from ENCODE[38]. Data were accessed from the UCSC Genome Browser[58] and the Gene Expression Omnibus (http://www.ncbi.nlm.nih.gov/geo/). Hi-C data in a B-cell line (GM12878)[39–42], including the contact matrix heatmap and virtual 4C results, were adapted from the YUE lab Hi-C Interactions and Virtual 4 C website (http://www.3dgenome.org, manuscript submitted for publication).

**Functional evaluation of *ARID5B* using in vitro models**. To examine the functional role of ARID5B, we used small interfering RNA (siRNA)-mediated *ARID5B* knockdown in LPS-stimulated human THP1-monocytes. Using the protocols as described above, we performed transcriptomic profiling using the Illumina HumanHT-12 v4 Expression BeadChip and measured *ARID5B* mRNA expression by quantitative PCR.

Human monocytic THP-1 cell line was purchased from the American Tissue Culture Collection. Cells were maintained in complete RPMI 1640 medium (Invitrogen) supplemented with 100 units per ml of penicillin, 100 µg ml$^{-1}$ of streptomycin, 2 mM L-glutamine, and 10% fetal bovine serum (FBS, HyClone, Logan, UT) in a humidified incubator with 5% $CO_2$ at 37 °C.

Two target-specific siRNAs for ARID5B exon 8&9 (Life Technologies, siRNA ID: s38579, designated as siARID5B 1 in Supplementary Fig. 4A) and for exon 6 (siRNA ID: s38580, designated as siARID5B 2 in Supplementary Fig. 4A) were used initially to compare for the knockdown efficiency. Then, the siARID5B 1 (s38579) was chosen for the remaining in vitro experiments. SiRNA transfection was performed using THP1 monocytes that were split 24 h before transfection. 10 nM of siARID5B were electronically transfected into THP-1 cells for 24 and 48 h using Amaxa Human Monocyte Nucleofector Kit and an Amaxa nucleofector II device (Lonza, Inc.). Scrambled siRNAs were transfected as negative controls. Samples were incubated for 24 h, followed by 3 h of LPS (LPS: 100 ng ml$^{-1}$) stimulation.

Levels of human ARID5B and cytokines presented in Supplementary Fig. 5 were measured by quantitative real-time reverse transcriptase-PCR (RT-PCR) using gene-specific TaqMan probe sets in an ABI prism 7000 sequence detection system (Life Technologies). Glyceraldehyde 3-phosphate dehydrogense mRNA was the internal loading control.

To reduce false-positive rates, two independent experiments (four siARID5B vs. four scrambled siRNA samples per experiment) were performed. To detect differential expression between two groups with small sample sizes, the regularized t-test implemented in the limma R package was used[59]. For the differentially expressed genes, IPA (QIAGEN, Redwood City, www.qiagen.com/ingenuity) was used to examine the enrichment of canonical pathways and biofunctions.

To further interrogate the potential pro-inflammatory effect that ARID5B may have, we repeated the third siARID5B knockdown experiment as described above and performed ELISA assays of culture media for IL-1a, the pro-inflammatory cytokine that was most significantly reduced by ARID5B mRNA Knockdown. Supernatants collected from LPS stimulated THP1 cells were analyzed for human IL-1a production using commercial sandwich ELISA kit (R&D Systems) according to the manufacturer's instructions. The results are representative of three or more experiments performed in triplicated as means + SEM. Levels of IL-1a mRNA expression presented in Fig. 5b were measured by RT-PCR as described previously.

To further examine the effects of ARID5B knockdown on cellular functions suggested by the transcriptomic profile changes, we repeated the siARID5B knockdown experiment as described above and performed THP1-monoctye migration and phagocytosis assays. Cell migration was evaluated using Transwell inserts (6.5 mm diameter) with polycarbonate filters (5-µm pore size, Corning Costar) in 24-well plates. THP1 cells (30,000 cells per well) were added to the upper chamber of the insert. The lower chamber contained 600 µl of RPMI 1640 medium/1% FBS with chemokine, MCP-1 (40 ng ml$^{-1}$). The plates were incubated at 37 °C in 5% $CO_2$ for 24 h and cells that had migrated into the lower chamber were counted using cell countess. Cell phagocytosis was evaluated using yellow–green carboxylate-modified microspheres (Thermo Fisher Scientific) in 24-well plates. Cells (500,000 cells per well) were incubated with 2 µl of beads (beads:cells = 5:1) for 60 min. After thoroughly washing 10 times with cold PBS, % engulfed beads in cells were calculated using fluorescence-activated cell sorting.

**Data availability**. Genome-wide gene expression and DNA methylation data have been deposited in the NCBI Gene Expression Omnibus (GEO) and is accessible through GEO Series accession number GSE56047 and GSE56046, respectively.

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

## Acknowledgements

This research was supported by contracts N01-HC-95159, N01-HC-95160, N01-HC-95161, N01-HC-95162, N01-HC-95163, N01-HC-95164, N01-HC-95165, N01-HC-95166, N01-HC-95167, N01-HC-95168, and N01-HC-95169, and R01 HL 119962 from the National Heart, Lung, and Blood Institute. The MESA Epigenomics and Transcriptomics Studies were funded by R01HL101250, R01 DK103531-01, R01 DK103531, R01 AG054474, and R01 HL135009-01 to Wake Forest University Health Sciences. The research described in this publication was funded in part by the U.S. Environmental Protection Agency through RD831697 to the University of Washington (MESA Air); it has not been subjected to the Agency's required peer and policy review and therefore does not necessarily reflect the views of the Agency and no official endorsement should be inferred.

## Author contributions

Conceptualization: Y.L., J.D., D.H., R.P.T., C.M., J.H.S. Methodology: Y.L., I.H., J.D., R.P.T., C.E.M., J.H.S., L.M.R. Investigation: Y.L., L.M.R., J.D., L.H., K.L., T.Y., W.C., M. W., H.G.S., J.P., D.S., L.H., B.M.P., S.S.R, J.I.R., J.D.K., G.L.B., D.R.J., W.P., I.H., D.A.B., R.P.T., D.H., C.E.M., J.H.S. Writing, original draft: Y.L., L.M.R., J.D., R.P.T., C.M., J.H.S. Writing, review and editing: Y.L., L.M.R., J.D., M.W., H.G.S., J.P., D.S., L.H., B.M.P., S.S.R, J.I.R., J.D.K., G.L.B., D.R.J., W.P., I.H., D.A.B., D.H., R.P.T., C.E.M., J.H.S. Visualization: Y.L., L.M.R., K.L. Funding acquisition, Y.L., I.H., J.D., D.H., R.P.T., C.E.M., J.H.S. Supervision: Y.L.

## Additional information

**Competing interests:** The authors declare no competing financial interests.

