## [Peer Review File · Nature Communications]

Reviewers' comments:

Reviewer #1 (Remarks to the Author):

This manuscript reports an interesting and novel finding that ARID5b is a gene significantly overexpressed at the mRNA level and hypomethylated in its enhancer region in sub-clinical atherosclerosis patient's blood monocytes. For this study, the authors carried out transcriptomics and methylomics studies using CD14+ blood monocytes by array studies, followed by systems biological analyses. An in vitro study using THP1 cells further showed that ARID5b knockdown had a widespread significant effects on hundreds of genes that regulate several pathways including inflammatory pathways. Overall, this is an interesting study that has been well-performed and the results provide supports for most conclusions drawn.

There are few concerns, however.

1. Array studies need additional validation using independent approaches. Given the importance of ARID5b, it is critical that additional data by qPCR and western blot shown in representative patient samples.
2. The functional study using the THP1 cell line with LPS as stimulant should be expanded by using primary monocytes and a relevant pro-atherogenic stimulus (e.g. TNF α , oxLDL etc) should be carried out to show its relevance in atherosclerosis.
3. ARID5b CpG methylation and its correlation with mRNA expression is only modest ($r=-0.21$) (Figure 3a). Additional studies testing that whether the ARID5b CpG methylation occurs in some human monocytes by independent means (e.g. bisulfite sequencing). Also, ARID5b transcription studies using a mutant in the CpG site will be definitive.

Reviewer #2 (Remarks to the Author):

The authors have studied monocytes from subjects in MESA and performed transcriptional and methylation profiling using Illumina array technology. This was then related to the clinical phenotype, in particular to coronary artery calcium (CAC) and carotid plaque score (the number of carotid plaques). Among the key findings was the identification of ARID5B as being associated with the severity of atherosclerotic disease at the transcriptional level, and also apparently via the methylation status of this gene. As important positives, these data come from a particularly important and well characterized clinical cohort of subjects. While the study appears well conducted and the manuscript is of interest, I have the following comments/suggestions:

Major

1. I find the use of the terms subclinical CVD and clinical CVD to be inconsistent and problematic. First, the terminology used for "clinical CVD" is inconsistent. On page 10 it is called both "clinical CVD" and "prevalent CVD", while in Figure 4 it is just called "CVD". Second, it is not stated if patients who suffered a clinical event were removed from the subclinical analysis. These patients have obviously suffered a CVD-related event, so by definition they cannot have subclinical disease. Third, given that both CAC and carotid plaque score were determined, for the purposes of this study if the authors want to use the term subclinical CVD it would make more sense for it to be defined as any carotid plaque or any CAC score > 0 in a patient with no prior CVD event. As an alternative, I believe it would be more accurate and clear if rather than using the term "subclinical CVD", the term "presence of any carotid plaque" is used.

2. Related to the above, it would be helpful to show the distribution histograms for CAC and carotid plaque score in the supplemental data. At present there is almost no information given on the burden of disease in these patients. Adding it to the supplemental data would be appropriate.
3. In figure 4, CAC is not included. All the other analyses to this point were for both CAC and carotid disease score. Its absence from figure 4 is unexpected. Depending on the distribution of the CAC data, there could be an arbitrary cut off use of a CAC score of 0, or say 100 for the purposes of the analysis in Figure 4.
4. The in vitro validation for ARID5B is encouraging, but not strong enough in my opinion. Figure 5a should be supplemental. Figure 5C is fairly uninformative. It looks dramatic juxtaposing the red against the blue, but it gives no indication of fold-change. Furthermore, how were the genes in Figure 5c selected? It appears to have been arbitrary and therefore possibly the most dramatically different genes have been presented. It would be more objective to show a heat map that includes all the genes for inflammation or inflammatory response from a publically recognized gene list, and to clearly highlight those where the difference was statistically significant. In addition, to further strengthen the argument of a pro-inflammatory monocyte/macrophage phenotype, profiling of control and knockdown cells using FACS showing a shift to a more inflammatory state would be more convincing. This could be combined with in vitro phenotypic profiling of the control and knockdown cells (migration, invasion, cytokine production - there are many possible in vitro assays). Such in vitro studies are not too onerous especially as the knockdown in THP-1 cells has already been successfully done. Of course, in vivo studies would be better still, but I appreciate this may be beyond the intended scope of this current submission.

Minor

1. Table 1 legend - define JHU +CO and UMN + WFU
2. Table 1 - for carotid plaque score and other measures where the range is given, it would be more informative to use a measure of statistical distribution such as interquartile range.
3. P8 line 3. Is 'replicating' the correct word here? This is not a true replication, just consistency of effect seen between study sites.
4. P9 - in the text the authors refer to Figure 3D. I presume this is 3C?
5. P 10, regarding this statement: "To examine the dose response relationship between ARID5B and extent of atherosclerosis, which may indicate their potential contribution to the progression of plaques, we performed linear regression analysis while excluding those with zero value of plaque score. The associations with carotid plaque score remain significant for the ARID5B mRNA expression ($p=3.4 \times 10^{-3}$) and cg25953130 methylation ($p=8.2 \times 10^{-3}$)." In order to demonstrate a "dose response effect" the authors need to quote the r or r2 value, and preferably show the correlation as a figure/supp figure. A p value gives no indication of the dose-effect.
6. LDL-C needs to be added to Table S7 and S8.
7. P 11, second para. In this entire para there is no data presented at all. This is pure discussion and should be moved to that section of the paper (or deleted).

Reviewer #3 (Remarks to the Author):

The manuscript entitled "Transcriptomics and Methylomics of Atherosclerosis in Human Blood Monocytes - The Multi-Ethnic Study of Atherosclerosis (MESA)" by Liu and colleagues studied RNA transcription and DNA methylation signatures associated with atherosclerosis and show that ARID5B expression can mediate cg25953130 methylation on atherosclerotic burden. The role of methylation on common human diseases is of great interest. The main sticking point in the field is trying to understand mechanistically how methylation alterations can cause increase risk of disease. Through integrating genome-wide transcription and methylation data from monocytes, together with other

published epigenetic signatures from DHSs, Hi-C and CHIA-PET, the authors provide a potential causal chain of ARID5B methylation -> ARID5B expression -> atherosclerosis. From this point, this manuscript is very exciting. However, I have some concerns, which need to be addressed:

Major comments:

1. The author claimed that "Our data ... strongly support the presence of an ARID5B regulatory region in the ARID5B gene body flanking ARID5B cg25953130". Even though mediation analysis and in Silico analyses can, in part, support this conclusion, it is important to validate this with additional biochemistry experiments, considering this is the main point of the manuscript. The authors should show that alteration in the methylation level of cg25953130 (eg. 5-aza-dC treatment, or clinical samples with different methylation level) in monocytes can affect the formation of 3D chromatin structure, or the binding of transcription factors (eg. EP300), and then ARID5B expression.
2. It has been known that there are some "problematic" probes in the 450K array, that can affect the measurement of methylation levels (eg. probes that can bind to multiple locations in the genome; probes with SNPs in the binding sites, etc.). Moreover, the 450K array only covers a small number of CpGs per gene region. The authors should do some direct validations and local fine-mapping of the methylation patterns for ARID5B cg25953130 loci that they discuss, using bisulfite sequencing (with pyrosequencing, or cloning and Sanger or Hiseq) on several monocytes.
3. It is interesting that the authors showed that the associations of ARID5B expression and methylation with carotid plaque and CAC are specific in CD14+ monocytes, but not in CD4+ T cells. However, the authors did not address whether there is still an inverse correlation between ARID5B methylation and expression in T cells. If not, I am curious to know whether the formation of chromatin structure, or transcription factor bindings has been affected in T cells.

Other comments:

1. Figure S1 is not mentioned in the text.
2. Top of page 6: "Additionally adjustments in the full model" should be "Additional adjustments..."
3. Legends in Table S7: "four genes bolded had genome-wide..." should be "three genes bolded..."
4. Are the pathways showed in Fig. 5B and Table S10 most significant ones? If so, this needs to be indicated in the text. If not, the full list needs to be provided.
5. Middle of page 13: "In addition, when examining decreased and increased genes separately, we identified regulation of transcription and ribonucleoprotein complex biogenesis genes to be enriched among up regulated genes (enrichment FDR<0.05, Table S10)." This seems wrong.

Reviewers' comments:

Reviewer #2 (Remarks to the Author):

The authors have submitted a much improved manuscript and addressed most of the comments and concerns. From my perspective, the single outstanding issue is the adequacy of the functional in vitro validation of the role/effect of ARID5B. In addition to myself (my original comment #4), I note Reviewer 1 also had the same concern (comment #2). Despite the fact that we both suggested this, all that was added to this revision was quantitation of IL-1a levels with ARID5B knockdown in THP-1 cells. I believe more extensive functional in vitro studies should be undertaken as originally suggested. Indeed, in the discussion the authors state that "The ARID5B regulated genes are expected to increase the activation state of atherosclerosis-relevant functions, including leukocyte chemotaxis, migration, extravasation, and phagocytosis, as well as lipid synthesis." This is therefore a major conclusion of this paper, and I believe that rather than speculation, proof of some of these effects needs to be demonstrated.

Reviewer #3 (Remarks to the Author):

I have reviewed the revised manuscript, "Transcriptomics and Methyloomics of Atherosclerosis in Human Blood Monocytes – The Multi-Ethnic Study of Atherosclerosis (MESA)", by Liu et al. The authors have adequately responded to my requests by clarification of the potential causal role of methylation, and providing the validation of methylation data using bisulfite sequencing.

One specific comment:

It seems that the disease status (T2D and CAC) for each individual is not available in the GEO deposit (GSE56047) provided by the authors. Full access to data is required by the policy of Nature Communications and will be of wide interest to the community.

REVIEWERS' COMMENTS:

Reviewer #2 (Remarks to the Author):

The authors have submitted a revised version of their manuscript, for which they conducted additional in vitro studies of THP-1 cells with knockdown of ARID5B. These studies showed a relevant cell phenotype with reduced migration and also marginally reduced phagocytosis. This satisfactorily addresses my prior concerns. I congratulate the authors on a very solid manuscript.

REVIEWER'S COMMENTS TO AUTHORS:

Reviewer: 1

1. Array studies need additional validation using independent approaches. Given the importance of ARID5b, it is critical that additional data by qPCR and western blot shown in representative patient samples.

To validate the microarray-based association results, we have added results with ARID5B expression measured using RNA-sequencing in a subset of 354 samples, which provide additional evidence for the associations between ARID5B expression and CAC ($\beta \pm SE = 0.45 \pm 0.20$, $p = 0.03$) and plaque ($\beta \pm SE = 0.13 \pm 0.06$, $p = 0.04$). These results have been added to page 6, 1st paragraph, last sentence.

To measure ARID5B protein levels, we have attempted to use western blot, however, the experiments have been unsuccessful. We are trying to figure out if this is due to the antibodies we are using, and will continue to pursue this line of investigation, but believe this is beyond the scope of this paper.

To further validate our methylation results, we performed pyrosequencing on DNA from 90 monocyte samples, which strongly correlate with the methylation results from microarray ($r=0.92$, $p=5.2 \times 10^{-37}$). We now report that ARID5B cg25953130 methylation using pyrosequencing significantly associated with carotid plaque ($\beta \pm se = -0.07 \pm 0.03$, $p=0.006$), and have added these results to page 8, 2nd paragraph, last sentence.

2. The functional study using the THP1 cell line with LPS as stimulant should be expanded by using primary monocytes and a relevant pro-atherogenic stimulus (e.g. TNF α , oxLDL etc) should be carried out to show its relevance in atherosclerosis.

We agree that additional functional experiments are needed to show the relevance of ARID5b in atherogenesis, as we acknowledge as a future direction in our discussion (page 16). However, primary monocytes cannot be transfected. We are funded for *in vivo* studies to investigate the role of ARID5b in atherogenesis.

3. ARID5b CpG methylation and its correlation with mRNA expression is only modest ($r=-0.21$) (Figure 3a). Additional studies testing that whether the ARID5b CpG methylation occurs in some human monocytes by independent means (e.g. bisulfite sequencing). Also, ARID5b transcription studies using a mutant in the CpG site will be definitive.

Epigenetic regulation has been thought to play a role in fine-tuning the expression levels of nearby genes, therefore, the observed modest correlation between ARID5b CpG methylation and its mRNA expression is likely to be reasonable. However, we have used an independent mean, pyrosequencing, and validated the ARID5b CpG methylation status for 90 monocyte samples as shown in the response to the question #1.

Whether the methylation change at this specific CpG is causal is highly debatable, given the likelihood that methylation change is caused by additional transcription factors binding to this enhancer region. Thus, mutating the CpG may not influence gene expression. We have added the following clarification in the discussion (page 15, 1st paragraph, 2nd sentence).

"It is worth noting that the observed inverse association of the *ARID5B* methylation site with its expression is not sufficient to support its causal role, since methylation change could be caused by additional transcription factors binding to this enhancer region."

Reviewer: 2

1. I find the use of the terms subclinical CVD and clinical CVD to be inconsistent and problematic. First, the terminology used for "clinical CVD" is inconsistent. On page 10 it is called both "clinical CVD" and "prevalent CVD", while in Figure 4 it is just called "CVD". Second, it is not stated if patients who suffered a clinical event were removed from the subclinical analysis. These patients have obviously suffered a CVD-related event, so by definition they cannot have subclinical disease. Third, given that both CAC and carotid plaque score were determined, for the purposes of this study if the authors want to use the term subclinical CVD it would make more sense for it to be defined as any carotid plaque or any CAC score > 0 in a patient with no prior CVD event. As an alternative, I believe it would be more accurate and clear if rather than using the term "subclinical CVD", the term "presence of any carotid plaque" is used.

We have added text to explicitly define prevalent CVD (page 10, 2nd paragraph, 1st sentence), and use the term consistently throughout the text. We have removed the term subclinical CVD, and added the term "presence of carotid plaque" instead (defined on page 10, 2nd paragraph, 1st sentence).

2. It would be helpful to show the distribution histograms for CAC and carotid plaque score in the supplemental data.

The distributions for CAC and carotid plaque score have been added to the supplemental information as Supplemental Figure 1 (Supplemental information, page 13).

3. In figure 4, CAC is not included. All the other analyses to this point were for both CAC and carotid disease score. Its absence from figure 4 is unexpected. Depending on the distribution of the CAC data, there could be an arbitrary cut off use of a CAC score of 0, or say 100 for the purposes of the analysis in Figure 4.

We have added text to define presence of CAC (page 10, 2nd paragraph, 1st sentence), and now include the results for presence of CAC in Figure 4.

4. The in vitro validation for *ARID5B* is encouraging, but not strong enough in my opinion. Figure 5a should be supplemental. Figure 5c is fairly uninformative. It looks dramatic juxtaposing the red against the blue, but it gives no indication of fold-change. Furthermore, how were the genes in Figure 5c selected? It appears to have been arbitrary and therefore possibly the most dramatically different genes have been presented. It would be more objective to show a heat map that includes all the genes for inflammation or inflammatory response from a publically recognized gene list, and to clearly highlight those where the difference was statistically significant. In addition, to

further strengthen the argument of a pro-inflammatory monocyte/macrophage phenotype, profiling of control and knockdown cells using FACS showing a shift to a more inflammatory state would be more convincing. This could be combined with in vitro phenotypic profiling of the control and knockdown cells (migration, invasion, cytokine production - there are many possible in vitro assays). Such in vitro studies are not too onerous especially as the knockdown in THP-1 cells has already been successfully done. Of course, in vivo studies would be better still, but I appreciate this may be beyond the intended scope of this current submission.

The fold-change and p values for all genes in the listed subsets of enriched categories were shown in Table S9, however, due to the number of genes in these categories, the heatmap would not be readable, therefore we subset out well known key genes to show as examples. We have moved Figure 5a and 5c to the supplemental figures (Figure S4, Supplemental information page 16).

Minor

1. Table 1 legend - define JHU +CO and UMN + WFU

We have added requested definitions to the Table 1 legend (page 27).

2. Table 1 - for carotid plaque score and other measures where the range is given, it would be more informative to use a measure of statistical distribution such as interquartile range.

We have updated Table 1 to include the interquartile range instead of range for these variables (page 27).

3. P8 line 3. Is 'replicating' the correct word here? This is not a true replication, just consistency of effect seen between study sites.

We have removed the word replicating.

4. P9 - in the text the authors refer to Figure 3D. I presume this is 3C?

We have corrected this typo, to refer to the correct figure, Figure 3C.

5. P 10, regarding this statement: "To examine the dose response relationship between ARID5B and extent of atherosclerosis, which may indicate their potential contribution to the progression of plaques, we performed linear regression analysis while excluding those with zero value of plaque score. The associations with carotid plaque score remain significant for the ARID5B mRNA expression ($p=3.4 \times 10^{-3}$) and cg25953130 methylation ($p=8.2 \times 10^{-3}$).\" In order to demonstrate a \"dose response effect\" the authors need to quote the r or r2 value, and preferably show the correlation as a figure/supp figure. A p value gives no indication of the dose-effect.

We have added the r2 values (see page 10, 2nd paragraph, last sentence).

6. LDL-C needs to be added to Table S7 and S8.

LDL-C was not associated with any atherosclerosis-associated gene expression and CpG methylation. We have added it in the footnotes for Table S7 and S8 (see Supplemental information, pages 29 and 31).

7. P 11, second para. In this entire para there is no data presented at all. This is pure discussion and should be moved to that section of the paper (or deleted).

We have deleted this paragraph.

Reviewer #3

Major comments:

1. The author claimed that "Our data ... strongly support the presence of an ARID5B regulatory region in the ARID5B gene body flanking ARID5B cg25953130". Even though mediation analysis and in Silico analyses can, in part, support this conclusion, it is important to validate this with additional biochemistry experiments, considering this is the main point of the manuscript. The authors should show that alteration in the methylation level of cg25953130 (eg. 5-aza-dC treatment, or clinical samples with different methylation level) in monocytes can affect the formation of 3D chromatin structure, or the binding of transcription factors (eg. EP300), and then ARID5B expression.

See response to Reviewer #1 comment #3

2. It has been known that there are some "problematic" probes in the 450K array, that can affect the measurement of methylation levels (eg. probes that can bind to multiple locations in the genome; probes with SNPs in the binding sites, etc.). Moreover, the 450K array only covers a small number of CpGs per gene region. The authors should do some direct validations and local fine-mapping of the methylation patterns for ARID5B cg25953130 loci that they discuss, using bisulfite sequencing (with pyrosequencing, or cloning and Sanger or Hiseq) on several monocytes.

See response to Reviewer #1 comment #1.

3. It is interesting that the authors showed that the associations of ARID5B expression and methylation with carotid plaque and CAC are specific in CD14+ monocytes, but not in CD4+ T cells. However, the authors did not address whether there is still an inverse correlation between ARID5B methylation and expression in T cells. If not, I am curious to know whether the formation of chromatin structure, or transcription factor bindings has been affected in T cells.

There was a significant inverse correlation between ARID5B methylation and expression in T cells ($r = -0.45$, $p = 1.27 \times 10^{-31}$), which we now report in results (page 11, last paragraph, 3rd sentence).

Other comments:

1. Figure S1 is not mentioned in the text.

Figure S1 was mentioned in the Methods, Study Design section on page 17. We have renamed this as Figure S5 to reflect its order as the last supplemental figure (see Supplemental information page 17).

2. Top of page 6: "Additionally adjustments in the full model" should be "Additional adjustments..."

Thank you, we have updated this typo.

3. Legends in Table S7: "four genes bolded had genome-wide..." should be "three genes bolded..."

We have updated the text to read three bold genes (Supplemental information, page 29).

4. Are the pathways showed in Fig. 5B and Table S10 most significant ones? If so, this needs to be indicated in the text. If not, the full list needs to be provided.

This is the full list, as indicated in the text (page 13, first sentence).

5. Middle of page 13: "In addition, when examining decreased and increased genes separately, we identified regulation of transcription and ribonucleoprotein complex biogenesis genes to be enriched among up regulated genes (enrichment $FDR < 0.05$, Table S10)." This seems wrong.

Agreed, the sentence shouldn't be there in the first place. We have deleted it.

RESPONSE TO REVIEWER'S COMMENTS TO AUTHORS:

Reviewer #2

1. I believe more extensive functional *in vitro* studies should be undertaken as originally suggested. Indeed, in the discussion the authors state that “The ARID5B regulated genes are expected to increase the activation state of atherosclerosis-relevant functions, including leukocyte chemotaxis, migration, extravasation, and phagocytosis, as well as lipid synthesis.” This is therefore a major conclusion of this paper, and I believe that rather than speculation, proof of some of these effects needs to be demonstrated.

We have performed additional *in vitro* functional validation. These results have been added to page 13, 3rd paragraph.

“To further examine the effects of *ARID5B* knockdown on cellular functions suggested by the transcriptomic profile changes, we performed THP1-monocyte migration and phagocytosis assays. *ARID5B* knockdown suppressed monocyte migration ($p=2.90 \times 10^{-5}$, 0.004 in experiment 1 and 2, respectively, **Figure 5C**). *ARID5B* knockdown also moderately inhibited monocyte phagocytosis ($p=0.008$) as shown in **Figure 5C**.”

Reviewer #3

1. Full access to data is required by the policy of Nature Communications and will be of wide interest to the community.

We agree that full access to the data will be of wide interest to the community. In order to add the disease status to the sample descriptions in GEO deposit GSE56047, we contacted GEO and sent the disease status updates (ticket #18411548). However, because the disease status information would become publicly available immediately following our request, we have canceled the first request and plan to provide the updated the sample descriptions including the disease status in a new request immediately following notification that the manuscript is accepted for publication.